

# Reviews and syntheses: Revisiting the boron systematics of aragonite and their application to coral calcification

Thomas M. DeCarlo[1,2], Michael Holcomb[1,2], and Malcolm T. McCulloch[1,2]

[1]Oceans Institute and School of Earth Sciences, The University of Western Australia, 35 Stirling Hwy, Crawley 6009, Australia
[2]ARC Centre of Excellence for Coral Reef Studies, The University of Western Australia, 35 Stirling Hwy, Crawley 6009, Australia

*Correspondence to:* Thomas M. DeCarlo (thomas.decarlo@uwa.edu.au)

**Abstract.** The isotopic and elemental systematics of boron in aragonitic coral skeletons have recently been developed as a proxy for the carbonate chemistry of the coral extracellular calcifying fluid. With knowledge of the boron isotopic fractionation in seawater and the B/Ca partition coefficient ($K_D$) between aragonite and seawater, measurements of coral skeleton $\delta^{11}$B and B/Ca can potentially constrain the full carbonate system. Two sets of abiogenic aragonite precipitation experiments designed to quantify $K_D$ have recently made possible the application of this proxy system. However, while different $K_D$ formulations have been proposed, there has not yet been a comprehensive analysis that considers both experimental datasets and explores the implications for interpreting coral skeletons. Here, we evaluate four potential $K_D$ formulations: three previously presented in the literature and one newly developed. We assess how well each formulation reconstructs the known fluid carbonate chemistry from the abiogenic experiments, and we evaluate the implications for deriving the carbonate chemistry of coral calcifying fluid. Three of the $K_D$ formulations performed similarly when applied to abiogenic aragonites precipitated from seawater and to coral skeletons. Critically, we find that some uncertainty remains in understanding the mechanism of boron elemental partitioning between aragonite and seawater, and addressing this question should be a target of additional abiogenic precipitation experiments. Despite this, boron systematics can already be applied to quantify the coral calcifying fluid carbonate system, although uncertainties associated with the proxy system should be carefully considered for each application. Finally, we present a user-friendly computer code that calculates coral calcifying fluid carbonate chemistry, including propagation of uncertainties, given inputs of boron systematics measured in coral skeleton.

## 1 Introduction

Quantifying the carbonate chemistry of the fluid from which corals accrete their skeletons is essential for understanding the mechanisms of skeletal growth and the sensitivity of skeletal composition to environmental variability. It is generally thought that corals precipitate aragonite ($CaCO_3$) crystals within an extracellular fluid-filled space between the living polyp and the skeleton (Barnes, 1970). Evidence from skeletal geochemistry and fluorescent dye experiments suggests that while seawater is the initial source of the calcifying fluid (McConnaughey, 1989; Adkins et al., 2003; Cohen and McConnaughey, 2003; Gagnon et al., 2012; Tambutté et al., 2012), the carbonate chemistry of the calcifying fluid is subject to substantial modifications (*i.e.*





pH and dissolved inorganic carbon, or DIC) that enhance the rapid nucleation and growth and aragonite crystals (Al-Horani et al., 2003; Venn et al., 2011). Because the isolation and small size of the calcifying fluid makes it difficult to sample directly, a variety of techniques have been employed to characterize its composition. These include microelectrodes inserted into tissue incisions or through the mouth (Al-Horani et al., 2003; Ries, 2011; Cai et al., 2016), pH-sensitive dyes (Venn et al., 2011, 2013; Holcomb et al., 2014; Comeau et al., 2017), Raman spectroscopy (DeCarlo et al., 2017), and a variety of skeletal-based geochemical proxies (Rollion-Bard et al., 2010, 2011; Inoue et al., 2011; Trotter et al., 2011; McCulloch et al., 2012b; Allison et al., 2014; Holcomb et al., 2014; DeCarlo et al., 2015). Although microelectrodes and pH-sensitive dyes are arguably the most direct methods, their utilities are limited by difficulties of applying them to corals living in their natural environment or developing seasonally-resolved time series. Geochemical proxies, although indirect, can be readily applied to the skeletons of corals living in both laboratory and natural environments, and to skeletons accreted years or even centuries ago.

In recent years, boron systematics (including $\delta^{11}$B and B/Ca) have become one of the most commonly applied proxies for the carbonate chemistry of coral calcifying fluid (cf) (Hönisch et al., 2004; Trotter et al., 2011; McCulloch et al., 2012b, a, 2017; Allison et al., 2014; DeCarlo et al., 2016; Stewart et al., 2016; Comeau et al., 2017; Wu et al., 2017; D'Olivo and McCulloch, 2017; Kubota et al., 2017; Ross et al., 2017; Schoepf et al., 2017). The sensitivity of boron isotopes to seawater pH arises from the borate versus boric acid speciation being pH-dependent and the isotopic fractionation between these species being constant (Klochko et al., 2006). Since the $\delta^{11}$B composition of aragonite precipitating from seawater reflects that of the borate species (Klochko et al., 2006; Trotter et al., 2011; Noireaux et al., 2015), the $\delta^{11}$B composition of the skeletal carbonate records the pH of the calcifying fluid. Furthermore, the B/Ca ratio depends inversely on the concentration of carbonate ion ($[CO_3^{2-}]$) since borate substitutes for carbonate ion in the aragonite lattice (Holcomb et al., 2016).

Use of combined boron isotopic ($\delta^{11}$B) and elemental (B/Ca) systematics has several advantages relative to other geochemical proxies. For example, while stable carbon and oxygen isotopes are sensitive to carbonate chemistry, their use is complicated by kinetic effects, strong sensitivities to the photosynthetic activity of coral symbionts, and variable compositions in seawater (Adkins et al., 2003; Cohen and McConnaughey, 2003; Schoepf et al., 2014). The U/Ca ratio of aragonite is also sensitive to $[CO_3^{2-}]$, but the amount of U in coral skeleton relative to its concentration in seawater suggests that $[U]_{cf}$ is depleted substantially, complicating its utility as a direct $[CO_3^{2-}]_{cf}$ proxy (DeCarlo et al., 2015). Conversely, the B/Ca and $\delta^{11}$B compositions of seawater are homogeneous (Foster et al., 2010; Lee et al., 2010) and likely not modified substantially by photosynthetic activity (Hönisch et al., 2004). Further, incorporation into the skeleton is less important for B/Ca than U/Ca because the partition coefficient between B and $[CO_3^{2-}]$ is at least 2 orders of magnitude smaller than that of $U/CO_3^{2-}$ (DeCarlo et al., 2015; Mavromatis et al., 2015; Holcomb et al., 2016), meaning that $[B]_{cf}$ is depleted much less than $[U]_{cf}$ as skeletal aragonite precipitates. While a low partition coefficient causes Rayleigh fractionation for elements in a closed system (*e.g.* coral $[Mg]/[Ca]_{cf}$) (Gaetani and Cohen, 2006), $[CO_3^{2-}]_{cf}$ is elevated relative to seawater and is modified by $CO_2$ diffusion and pH up-regulation (*i.e.* it is not in a closed system) (Adkins et al., 2003; Cai et al., 2016), meaning that $[B]/[CO_3^{2-}]_{cf}$ is likely not changed substantially due to skeletal aragonite precipitation. Therefore, boron-based proxies are thought to be largely dependent on carbonate chemistry alone (Trotter et al., 2011; McCulloch et al., 2017). Finally, the combination of two carbonate system proxies (pH and $[CO_3^{2-}]$) derived from boron systematics allows for computation of the full carbonate system (Zeebe and Wolf-Gladrow, 2001).





Abiogenic laboratory experiments provide the underlying quantitative foundation necessary to apply these proxies to aragonitic coral skeletons. Klochko et al. (2006) determined the fractionation factor ($\alpha_{B3-B4}$) between boric acid and borate in seawater, which allows $\delta^{11}$B of carbonates to be used as a pH proxy when combined with knowledge of p$K_B$ (Dickson, 1990) and seawater $\delta^{11}$B (Foster et al., 2010). Although there is potential for B isotopic fractionation between aragonite and seawater

(Balan et al., 2018), the veracity of the $\delta^{11}$B proxy has been largely confirmed by comparison with direct in-situ measurements using either pH micro-electrodes or confocal microscopy of pH-sensitive dyes in the calcifying fluid (Ries, 2011; Venn et al., 2011; Holcomb et al., 2014; Cai et al., 2016). Additionally, results from two sets of abiogenic precipitation experiments can be used to constrain the partitioning of B/Ca between fluid and aragonite (Mavromatis et al., 2015; Holcomb et al., 2016). Thus, while all the information theoretically required to constrain the full seawater carbonate system from boron systematics is now

available, a variety of different approaches have been presented, especially regarding the interpretation of B/Ca partitioning (Mavromatis et al., 2015; Holcomb et al., 2016; Allison, 2017; McCulloch et al., 2017). Here, we assess the abiogenic partitioning data (Mavromatis et al., 2015; Holcomb et al., 2016), and the subsequent fitting of those data (Allison, 2017; McCulloch et al., 2017). We consider which mechanisms of B incorporation and sensitivities of B/Ca partitioning are plausible, and the implications for interpreting coral skeletons. Finally, we present a user-friendly computer code to calculate coral calcifying

fluid carbonate chemistry from measurements of $\delta^{11}$B and B/Ca. The code also propagates known uncertainties for deriving calcifying fluid [CO$_3^{2-}$]$_{cf}$ and DIC$_{cf}$, and allows for evaluating the effects of using different constants and partition coefficient formulations.

## 2 Partitioning of B/Ca between aragonite and seawater

The main discrepancy among various applications of boron systematics to coral skeletons relates to the partition coefficient

of boron between aragonite and seawater. Given the variety of possible exchange reactions and partition coefficients that have been proposed (Allen and Hönisch, 2012; Mavromatis et al., 2015; Holcomb et al., 2016; Allison, 2017; McCulloch et al., 2017), we begin with a brief review of how partition coefficients are derived. In general, the substitution of minor elements into a solid is described by an exchange reaction such that:

$$X^{solid} + Y^{fluid} = Y^{solid} + X^{fluid} \tag{1}$$

For example, the substitution of Sr$^{2+}$ for Ca$^{2+}$ in aragonite follows (Gaetani and Cohen, 2006):

$$\text{Ca}^{aragonite} + \text{Sr}^{fluid} = \text{Sr}^{aragonite} + \text{Ca}^{fluid} \tag{2}$$

Element distribution described by this exchange is quantified through a partition coefficient, expressed as the concentration ratio of products over reactants:

$$K_D^{\text{Sr/Ca}} = \frac{[\text{Sr}]^{aragonite}[\text{Ca}]^{fluid}}{[\text{Sr}]^{fluid}[\text{Ca}]^{aragonite}} \tag{3}$$



Equation (3) is typically rearranged as:

$$K_D^{\text{Sr/Ca}} = \left( \frac{[\text{Sr}]^{aragonite}}{[\text{Sr}]^{fluid}} \right) \left( \frac{[\text{Ca}]^{fluid}}{[\text{Ca}]^{aragonite}} \right) = \left( \frac{[\text{Sr}]^{aragonite}}{[\text{Sr}]^{fluid}} \right) \left( \frac{[\text{Ca}]^{aragonite}}{[\text{Ca}]^{fluid}} \right)^{-1} = \frac{\text{Sr/Ca}^{aragonite}}{\text{Sr/Ca}^{fluid}} \qquad (4)$$

The case of $Sr^{2+}$ substituting for $Ca^{2+}$ is straightforward in that the exchange reaction (Eq. 2) is charge-balanced. Boron is more complicated because it is commonly thought that the singly charged $B(OH)_4^-$ is incorporated into aragonite in place of the doubly charged $CO_3^{2-}$ (Mavromatis et al., 2015; Noireaux et al., 2015). There are at least two possible exchange reactions for $B(OH)_4^-$ to substitute for $CO_3^{2-}$ that maintain charge balance:

$$0.5\text{CaCO}_3 + \text{B(OH)}_4^- \leftrightarrow \text{Ca}_{0.5}\text{B(OH)}_4 + 0.5\text{CO}_3^{2-} \qquad (5)$$

following Holcomb et al. (2016), or:

$$\text{CaCO}_3 + \text{B(OH)}_4^- \leftrightarrow \text{CaH}_3\text{BO}_4 + \text{H}^+ + \text{CO}_3^{2-} \qquad (6)$$

following McCulloch et al. (2017). The $K_D$ for Eq. (5) is:

$$K_D^{\text{B/Ca}} = \frac{[\text{B(OH)}_4^- / [\text{CO}_3^{2-}]^{0.5}]^{aragonite}}{[\text{B(OH)}_4^- / [\text{CO}_3^{2-}]^{0.5}]^{fluid}} = \frac{[\text{B/Ca}]^{aragonite}}{[\text{B(OH)}_4^- / [\text{CO}_3^{2-}]^{0.5}]^{fluid}} \qquad (7)$$

and for Eq. (6) is:

$$K_D^{\text{B/Ca}} = \frac{[\text{B(OH)}_4^- / \text{CO}_3^{2-}]^{aragonite}}{[\text{B(OH)}_4^- / \text{CO}_3^{2-}]^{fluid}} = \frac{[\text{B/Ca}]^{aragonite}}{[\text{B(OH)}_4^- / \text{CO}_3^{2-}]^{fluid}} \qquad (8)$$

where $[\text{CO}_3^{2-}]^{aragonite}$ is assumed equal to $[\text{Ca}^{2+}]^{aragonite}$, and Eq. (7) and Eq. (8) differ by whether or not the square root of $CO_3^{2-}$ is used. Since Eq. (6) includes $H^+$ in the products, this reaction implies that the $K_D$ may be pH-dependent (McCulloch et al., 2017). Incorporation of B into aragonite may also involve adsorption of $B(OH)_4^-$ onto crystal surfaces, incorporation at defect sites, or local charge balance by $Na^+$ (Balan et al., 2018).

Conversely, Allison et al. (2014) and Allison (2017) considered exchange reactions in which borate substitutes for bicarbonate ($HCO_3^-$), with the partition coefficient:

$$K_D^{\text{B/Ca}} = \frac{[\text{B/Ca}]^{aragonite}}{[\text{B(OH)}_4^- / \text{HCO}_3^-]^{fluid}} \qquad (9)$$

This approach resolves the issue of charge balance and would account for a $CO_3^{2-}$ reacting with $H^+$, thus removing the pH dependence expected from Eq. (8). However, Eq. (9) implies that aragonite forms via the reaction:

$$\text{Ca}^{2+} + \text{HCO}_3^- \leftrightarrow \text{CaCO}_3 + \text{H}^+ \qquad (10)$$

rather than:

$$\text{Ca}^{2+} + \text{CO}_3^{2-} \leftrightarrow \text{CaCO}_3 \qquad (11)$$




Whether aragonite precipitates via Eq. (10) or Eq. (11) is a testable hypothesis because the rate of the net forward reaction should depend on the concentrations of the reactants. Burton and Walter (1987) demonstrated that the rate of aragonite precipitation increases as a function of $\Omega_{Ar}$ (where $\Omega_{Ar} = [Ca^{2+}][CO_3^{2-}]/K_{sp}$) and temperature, although they did not explicitly consider the relationship between $[HCO_3^-]$ and precipitation rate. Holcomb et al. (2016) reported bulk precipitation rates

for aragonites precipitated from seawater with various $[CO_3^{2-}]$ and $[HCO_3^-]$, with independence between these two variables achieved by manipulating pH and DIC. While the bulk precipitation rates were not normalized to surface area as in Burton and Walter (1987), the experimental vessels used by Holcomb et al. (2016) were of consistent dimensions and material. Thus, the bulk precipitation rate data of Holcomb et al. (2016) should be comparable among their experiments, allowing us to evaluate between the reactions of Eq. (10) and Eq. (11). The aragonite precipitation rates reported by Holcomb et al. (2016) at 25 °C

are significantly correlated with both $[CO_3^{2-}]$ (r² = 0.56, p < 0.01) and $\Omega_{Ar}$ (r² = 0.62, p < 0.01) (Figure 1a,b). Experiments conducted at 20 °C, 33 °C, and 40 °C are consistent with this trend (Figure 1a,b), although we do not attempt to quantify any temperature effects since only two experiments were conducted at each temperature other than 25 °C. Conversely, there are no significant correlations between aragonite precipitation rate at 25 °C and either $[HCO_3^-]$ (r² = 0.00, p = 0.95) or $[Ca^{2+}][HCO_3^-]$ (r² = 0.01, p = 0.54) as would be expected based on Eq. (10). Other possibilities include precipitation reactions involving both

$CO_3^{2-}$ and $HCO_3^-$, or total DIC (Allison et al., 2014; Allison, 2017). However, there are no significant correlations between precipitation rate and either $[CO_3^{2-}]+[HCO_3^-]$ (r² = 0.01, p = 0.59) or DIC (r² = 0.01, p = 0.59) (Figure 1e,f). Together, these data lead us to conclude that aragonite precipitates from seawater via Eq. (11). Therefore, since B/Ca partition coefficients expressed with $[HCO_3^-]$ do not have a chemical reaction basis, we do not consider them further. Rather, we consider only the B/Ca partition coefficients that are based on borate substituting for $CO_3^{2-}$ (Eqs. 7-8).

## 20 3  Fitting the experimental B/Ca partitioning data

The second source of discrepancies between various applications of boron systematics to coral skeletons is the dependence of the $K_D$ on fluid chemistry. Holcomb et al. (2016) fit the $K_D$ as either a function of $[CO_3^{2-}]$ or $\Omega_{Ar}$, McCulloch et al. (2017) refit the Holcomb et al. (2016) data as a function of $[H^+]$, and Allison (2017) fit data from both Mavromatis et al. (2015) and Holcomb et al. (2016) as a function of $\Omega_{Ar}$.

At the outset, it is important to recognize that there are two key differences between the abiogenic experiments of Mavromatis et al. (2015) and Holcomb et al. (2016). Firstly, Mavromatis et al. (2015) precipitated aragonite from NaCl solutions, whereas Holcomb et al. (2016) used filtered seawater. Secondly, $[CO_3^{2-}]$ and $\Omega_{Ar}$ are lower in the experiments of Mavromatis et al. (2015) relative to Holcomb et al. (2016). Potentially as result of one or both of these differences, Mavromatis et al. (2015) found much lower $K_D$ values than Holcomb et al. (2016). Here, we consider four possible $K_D$ dependencies based on these

two experimental datasets (Figure 2).

The first two formulations assume that there are substantial compositional effects on B/Ca partitioning, and thus the offsets in $K_D$ between Mavromatis et al. (2015) and Holcomb et al. (2016) arise due to the use of NaCl versus seawater solutions, respectively (Figure 2a,b). If this is correct, the Holcomb et al. (2016) data are more appropriate for application to corals



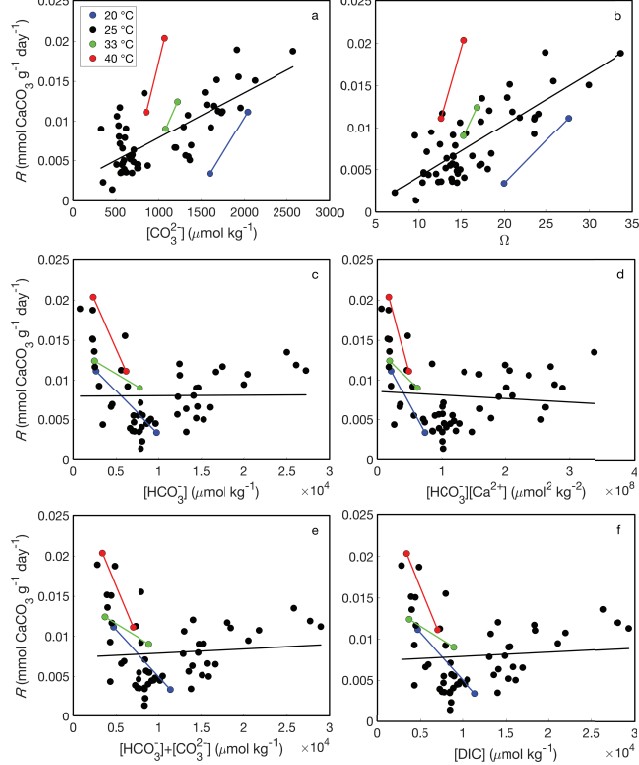

**Figure 1.** Aragonite precipitation rates as functions of fluid chemistry based on data from Holcomb et al. (2016). Each point represents a separate abiogenic aragonite precipitation experiment conducted at 20 °C (blue), 25 °C (black), 33 °C (green), and 40 °C (red). Bulk aragonite precipitation rates ($R$) are plotted against mean fluid $[CO_3^{2-}]$ (a), $\Omega_{Ar}$ (b), $[HCO_3^-]$ (c), $[Ca^{2+}][HCO_3^-]$ (d), $[CO_3^{2-}]+[HCO_3^-]$ (e), and DIC (f). Solid lines show regression fits at each temperature (note that there are only two experiments at each temperature other than 25 °C, and thus lines fit for these temperatures should be interpreted with caution).

based on evidence that they precipitate their skeletons from seawater-based solutions (McConnaughey, 1989; Cohen and Mc-Connaughey, 2003; Gagnon et al., 2012; Tambutté et al., 2012). We are then left with the two plausible $K_D$ expressions (Eqs. 7-8), and the previously presented dependencies on either $[CO_3^{2-}]$ (Holcomb et al., 2016) or $[H^+]$ McCulloch et al. (2017).

Alternatively, it is possible that there are negligible effects from using NaCl or seawater solutions and, therefore, the data

5   from both Mavromatis et al. (2015) and Holcomb et al. (2016) should be fit by a single, continuous function. There are again two plausible formulations: $K_D$ increases as a function of $[CO_3^{2-}]$ or $\Omega_{Ar}$ (Figure 2c,d). Allison (2017) proposed a linear fit between $K_D$ and $\Omega_{Ar}$ that includes both the Mavromatis et al. (2015) and Holcomb et al. (2016) data. From a practical standpoint, however, this latter approach is problematic in that it requires an independent proxy for $[Ca^{2+}]$ (see section 7) and the linear fit effectively precludes its use for deriving coral calcifying fluid chemistry (see section 4). In an attempt to avoid

10   these issues, we introduce a logarithmic relationship between $K_D$ and $[CO_3^{2-}]$, which fits both the Mavromatis et al. (2015)




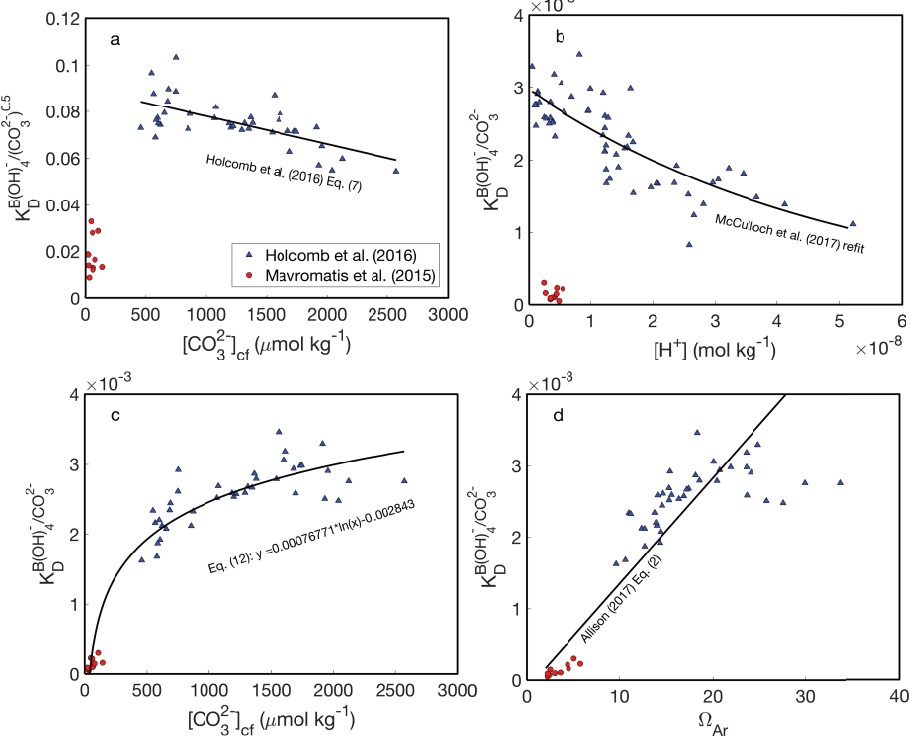

**Figure 2.** B/Ca $K_D$ formulations. Abiogenic B/Ca partitioning data from Mavromatis et al. (2015) (red circles) and Holcomb et al. (2016) (blue triangles) fit as functions of fluid chemistry: $[CO_3^{2-}]$ (a,c) (Holcomb et al., 2016), $[H^+]$ (b) (McCulloch et al., 2017), and $\Omega_{Ar}$ (d) (Allison, 2017). Note that $K_D$ in (a) is defined with Eq. (7) and in (b-d) is defined with Eq. (8). We use only the Mavromatis et al. (2015) with $[B] < 1000\ \mu$mol kg$^{-1}$ due to the apparent effect of $[B]$ on $K_D$ (Holcomb et al., 2016).

and Holcomb et al. (2016) data (Figure 2c):

$$K_D^{\text{B/Ca}} = 0.00077(\pm 0.00007) * ln([CO_3^{2-}]) - 0.0028(\pm 0.0004) \tag{12}$$

where parentheses indicate 95% confidence, $K_D$ is defined by Eq. (8), and $[CO_3^{2-}]$ is in units of $\mu$mol kg$^{-1}$. Mechanistically, the increase in $K_D$ with $[CO_3^{2-}]$ or $\Omega_{Ar}$ (or precipitation rate) is consistent with the surface entrapment model proposed by
5  Watson (2004). In this model, minor element impurities, such as B, are incorporated in the near-surface layer of a growing crystal. Slower growing crystals allow these impurities to diffuse out of the near-surface region into the fluid, whereas faster growing crystals bury the near-surface impurities into the bulk crystal. The sensitivity of $K_D$ to $[CO_3^{2-}]$ or $\Omega_{Ar}$ is also consistent with a surface kinetic model (DePaolo, 2011), in which trace element partitioning depends on the net rate of precipitation relative to dissolution. Thus, both the surface entrapment and kinetic models offer potential explanations as to why the low-$\Omega_{Ar}$
10  experiments of Mavromatis et al. (2015) produced lower $K_D$ than the higher-$\Omega_{Ar}$ experiments of Holcomb et al. (2016).





## 4 Back-application of partition coefficient formulations to abiogenic datasets

We conducted a simple test to evaluate the utility of the four $K_D$ dependencies considered above. For each $K_D$ formulation, we used the reported aragonite B/Ca, fluid $[B(OH)_4^-]$, and pH data of Mavromatis et al. (2015) and Holcomb et al. (2016) to calculate the fluid $[CO_3^{2-}]$, and then we compared the predicted $[CO_3^{2-}]$ to the concentrations measured during the experiments (Figure 3) (see also Ross et al. (2017) for a similar analysis). The basis for this approach is to assess how well the experimental fluid $[CO_3^{2-}]$ can be reconstructed using boron systematics alone. When boron systematics are applied to coral skeletons, $[CO_3^{2-}]$ is predicted from only B/Ca and $\delta^{11}B$. However, since $\delta^{11}B$ was not reported by Holcomb et al. (2016), we instead use the measured pH for the McCulloch et al. (2017) $K_D$ formulation. Additionally, since [B] was manipulated in some experiments, we use reported fluid $[B(OH)_4^-]$ instead of calculating it from pH as is done in applications to corals (Allison et al., 2014; McCulloch et al., 2017). Nevertheless, since pH (and thus seawater $[B(OH)_4^-]$) are readily calculated from $\delta^{11}B$, our approach is suitable for evaluating the utility of each $K_D$ formulation for reconstructing $[CO_3^{2-}]$ with B/Ca.

Since three of the $K_D$ formulations (Holcomb et al. (2016), Allison (2017), and our new Eq. 12) themselves depend on $[CO_3^{2-}]$, we solved for $[CO_3^{2-}]$ as follows. An initial guess of $[CO_3^{2-}]$ was used to calculate an initial $K_D$, and this $K_D$ was used to solve for $[CO_3^{2-}]$ by rearranging Eq. (7) to:

$$[CO_3^{2-}] = \left( K_D^{B/Ca} \frac{[B(OH)_4^-]^{fluid}}{[B/Ca]^{aragonite}} \right)^2 \tag{13}$$

and Eq. (8) to:

$$[CO_3^{2-}] = K_D^{B/Ca} \frac{[B(OH)_4^-]^{fluid}}{[B/Ca]^{aragonite}} \tag{14}$$

where Eq. (14) is used for Allison (2017) and our new Eq. (12), and Eq. (13) is used for Holcomb et al. (2016). We then calculated the residual between the calculated (Eqs. 13-14) and initially estimated $[CO_3^{2-}]$. Finally, we iteratively adjusted the initial $[CO_3^{2-}]$ estimate for each data point until it equaled the $[CO_3^{2-}]$ derived from Eqs. (13-14).

Both the Holcomb et al. (2016) fit (their equation 7) and the McCulloch et al. (2017) refit perform similarly, effectively reconstructing the fluid $[CO_3^{2-}]$ of the Holcomb et al. (2016) experimental data (root mean square error, RMSE = 151 and 163 $\mu$mol kg$^{-1}$, respectively), but performing poorly for the Mavromatis et al. (2015) data (RMSE = 1370 and 1385 $\mu$mol kg$^{-1}$, respectively) (Figure 3a,b). This is not surprising because these $K_D$ dependencies are offset from the Mavromatis et al. (2015) data (Figure 2a,b). Our new logarithmic equation performs well for both datasets (RMSE = 204 and 42 $\mu$mol kg$^{-1}$ for Mavromatis et al. (2015) and Holcomb et al. (2016), respectively). The Allison (2017) formulation (assuming $[Ca^{2+}]$ of 10 mmol kg$^{-1}$) performs well for the Mavromatis et al. (2015) data (RMSE = 51 $\mu$mol kg$^{-1}$), but creates a trend opposite that expected for the Holcomb et al. (2016) data (RMSE = 1375 $\mu$mol kg$^{-1}$) (Figure 3d). Using the reported $[Ca^{2+}]$ and $K_{sp}$ from the Holcomb et al. (2016) in the Allison (2017) formulation improves the results slightly and generates more positive solutions, but the RMSE is still 950 $\mu$mol kg$^{-1}$.

An alternative way to understand these patterns is to investigate the relationship between $[CO_3^{2-}]$ and the ratio of fluid $[B(OH)_4^-]$ to solid B/Ca (Figure 4). Following Eqs. (13-14), $[CO_3^{2-}]$ should be positively related to $\frac{[B(OH)_4^-]^{fluid}}{[B/Ca]^{aragonite}}$, and this behavior is clearly evident in the abiogenic aragonites of Holcomb et al. (2016) (blue triangles in Figure 4). The $K_D$ formulations





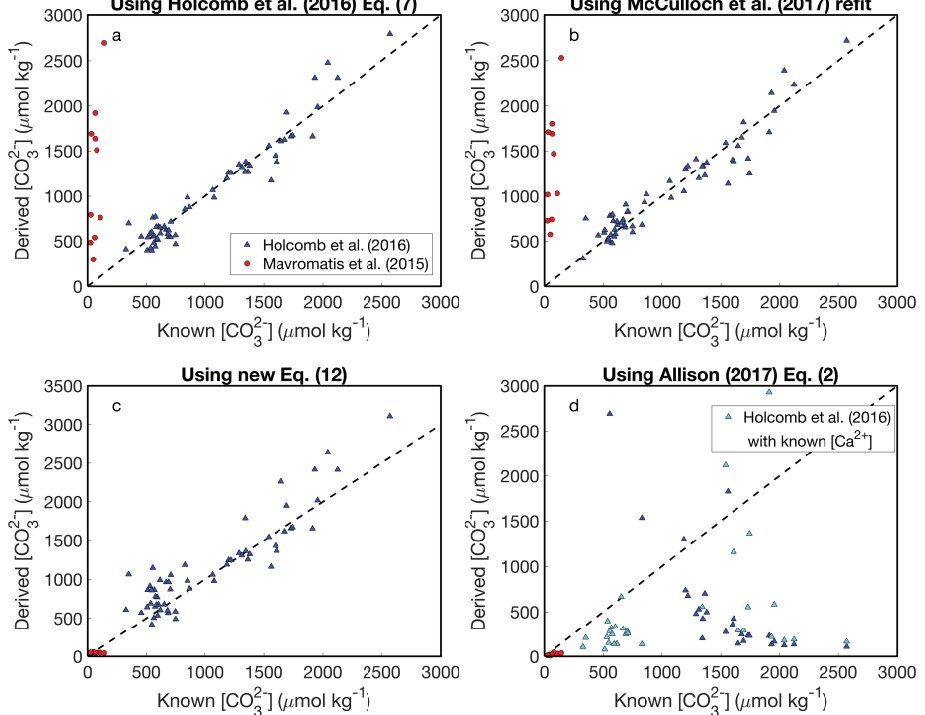

**Figure 3.** Reconstructing experimental fluid $[CO_3^{2-}]$ using the $K_D$ formulations presented in Figure 2. Symbols are the same as Figure 2. In panel (d), negative $[CO_3^{2-}]$ solutions have been excluded (see Appendix). Calculations using the Allison (2017) $K_D$ formulation have been performed with both assuming seawater $[Ca^{2+}]$ (blue) and using the $[Ca^{2+}]$ reported from the experiments (cyan).

of Holcomb et al. (2016), McCulloch et al. (2017), and our new Eq. (12) all closely track the abiogenic data, especially for $[CO_3^{2-}] < 2000\ \mu mol\ kg^{-1}$. Conversely, the Allison (2017) fit (assuming $[Ca^{2+}]$ of 10 mmol kg$^{-1}$) produces the opposite trend and is invalid or negative below a $\frac{[B(OH)_4^-]^{fluid}}{[B/Ca]^{aragonite}}$ of ~0.44 mol kg$^{-1}$ (see Appendix for derivation of an analytical solution).

The behavior of the $K_D$ formulations can be understood by inspecting the residuals between initial $[CO_3^{2-}]$ estimates and
5   those derived from Eqs. (13-14) (Figure 5). The Holcomb et al. (2016) $K_D$ formulation generates unique $[CO_3^{2-}]$ solutions (*i.e.* where the residual equals zero) that increase with $\frac{[B(OH)_4^-]^{fluid}}{[B/Ca]^{aragonite}}$ (Figure 5a), which is the ideal behavior. Our new Eq. (12) also produces increasing $[CO_3^{2-}]$ solutions with increasing $\frac{[B(OH)_4^-]^{fluid}}{[B/Ca]^{aragonite}}$ (Figure 5b), however, a major issue of this formulation is that there may be two $[CO_3^{2-}]$ solutions for each $\frac{[B(OH)_4^-]^{fluid}}{[B/Ca]^{aragonite}}$. Finally, although the Allison (2017) $K_D$ formulation produces unique $[CO_3^{2-}]$ solutions, they increase with decreasing $\frac{[B(OH)_4^-]^{fluid}}{[B/Ca]^{aragonite}}$ (Figure 5c), opposite to that expected (Figure 4).

10   The reason for the poor behavior of the Allison (2017) formulation is the linear fit between $K_D$ and $\Omega_{Ar}$ with an intercept near the origin. When using this formulation to predict $[CO_3^{2-}]$ from boron systematics alone, we must assume $[Ca^{2+}]$ is approximately equal to seawater (~10 mmol kg$^{-1}$), meaning that $\Omega_{Ar}$ is directly related to $[CO_3^{2-}]$. Since the intercept in the Allison (2017) $K_D$ formulation is close to the origin, any change in $[CO_3^{2-}]$ results in an almost proportional change in $K_D$. It can be seen why this is problematic by inspecting how $[CO_3^{2-}]$ is derived from Eq. (14). The $\frac{[B(OH)_4^-]^{fluid}}{[B/Ca]^{aragonite}}$ is derived



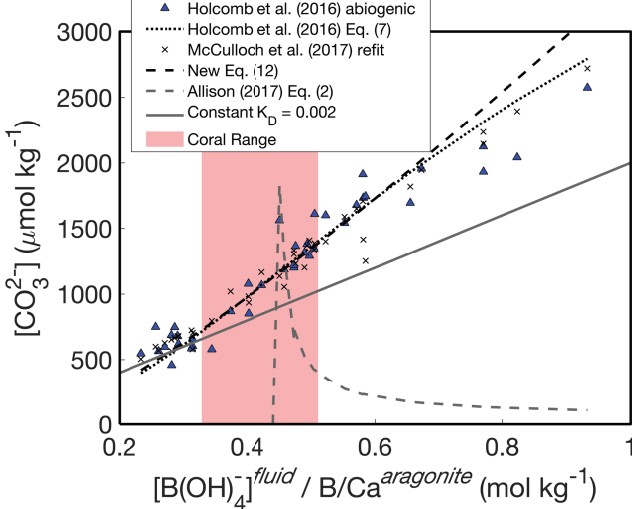

**Figure 4.** Experimental fluid $[CO_3^{2-}]$ as a function of $\frac{[B(OH)_4^-]^{fluid}}{[B/Ca]^{aragonite}}$. The $K_D$ formulations of Holcomb et al. (2016) (dotted black line), McCulloch et al. (2017) (black crosses), and Eq. (12) (dashed black line) all correctly capture the trend of increasing $[CO_3^{2-}]$ with increasing $\frac{[B(OH)_4^-]^{fluid}}{[B/Ca]^{aragonite}}$ that is apparent in the abiogenic data (blue triangles), whereas the Allison (2017) (dashed grey line) produces an opposing pattern. A constant $K_D$ (solid grey line) underestimates the slope between $\frac{[B(OH)_4^-]^{fluid}}{[B/Ca]^{aragonite}}$ and $[CO_3^{2-}]$. The pink shaded region indicates the range of $\frac{[B(OH)_4^-]^{fluid}}{[B/Ca]^{aragonite}}$ derived for *Porites* corals by McCulloch et al. (2017).

from pH (or $\delta^{11}B$) and measured B/Ca, so this ratio remains constant while we find the appropriate $K_D$ that minimizes the residual $[CO_3^{2-}]$, as in Figure 5. Therefore, Eq. (14) is effectively reduced to $[CO_3^{2-}]$ being a function of $K_D$ multiplied by a constant. However, since $K_D$ changes almost directly proportional to $[CO_3^{2-}]$ according to Allison (2017), it is difficult to find a $[CO_3^{2-}]$ that explains different $\frac{[B(OH)_4^-]^{fluid}}{[B/Ca]^{aragonite}}$. Although Allison (2017) recognized the difficulty of explaining the range of B/Ca observed in corals (see their Figure 8g), the implication of applying this $K_D$ formulation to predict $[CO_3^{2-}]$ was not

discussed. Our analysis suggests that this $K_D$ formulation is poorly suited for accurately reconstructing fluid $[CO_3^{2-}]$ from boron systematics (Figure 3d, Figure 4).

Another approach presented by Allison (2017) is to use a constant $K_D$. While a constant $K_D$ performs better than the linear fit to $\Omega_{Ar}$, it underestimates the slope of the relationship between $\frac{[B(OH)_4^-]^{fluid}}{[B/Ca]^{aragonite}}$ and $[CO_3^{2-}]$ (Figure 4). This is not surprising

because the abiogenic data clearly show the $K_D$ does not remain constant as $[CO_3^{2-}]$ changes (Figure 2). Since using a constant $K_D$ will underestimate variability in $[CO_3^{2-}]_{cf}$ when applied to corals, we do not recommend this approach.

## 5  Application to deriving coral calcifying fluid carbonate chemistry

The ability of boron systematics to predict two independent carbonate chemistry parameters allows for calculation of the full seawater carbonate system. This has prompted several recent applications deriving the carbonate chemistry of coral calcifying

fluids (Allison et al., 2014; Comeau et al., 2017; D'Olivo and McCulloch, 2017; Kubota et al., 2017; McCulloch et al., 2017;





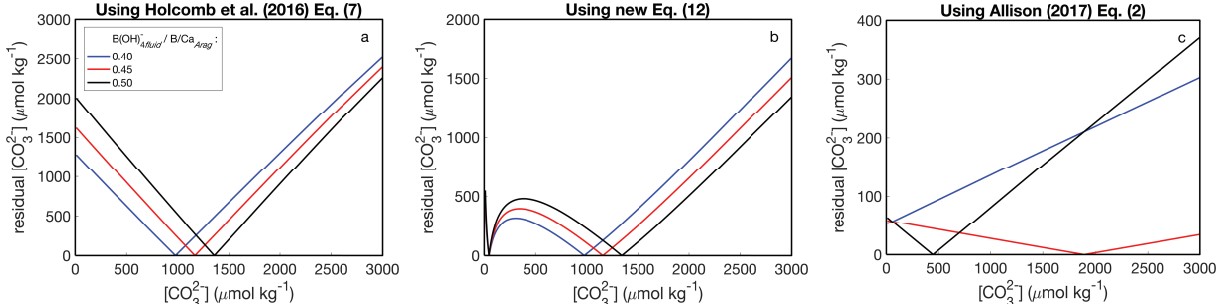

**Figure 5.** Predicting $[CO_3^{2-}]$ from the $K_D$ formulations which themselves depend on $[CO_3^{2-}]$: Holcomb et al. (2016) (a), Eq. (12) (b), and Allison (2017) (c). Each panel shows the residual between a guess of $[CO_3^{2-}]$ used to calculate $K_D$ and that calculated from Eqs. (13-14), plotted against the $[CO_3^{2-}]$ guess. The final $[CO_3^{2-}]$ is derived by finding where the residual is minimized for a particular $\frac{[B(OH)_4^-]^{fluid}}{[B/Ca]^{aragonite}}$ (three of which are plotted as examples in red, blue, and black).

Ross et al., 2017; Schoepf et al., 2017). Here, we investigate the differences in derived coral calcifying fluid $[CO_3^{2-}]$ and DIC that arise from the choice of $K_D$ formulation. We use the paired $\delta^{11}B$ and B/Ca data of the "Davies 2" coral from McCulloch et al. (2017) as an example.

Derived $[CO_3^{2-}]_{cf}$ and $DIC_{cf}$ show similar seasonality when using the $K_D$ formulations of Holcomb et al. (2016), McCul-
loch et al. (2017), or our new Eq. (12) (Figure 6). Regardless of which of these three $K_D$ formulations are used, $[CO_3^{2-}]_{cf}$ and $DIC_{cf}$ are both highest in summer and lowest in winter over a multi-year time series. This is consistent with an independent approach based on Rayleigh modelling of minor elements in coral skeleton (Gaetani and Cohen, 2006; Gaetani et al., 2011). The primary difference among the derived values is that the $K_D$ formulations from Holcomb et al. (2016) and our Eq. (12) produce seasonal cycles with ~50% greater amplitude relative to the McCulloch et al. (2017) $K_D$ formulation. The absolute
values of derived $[CO_3^{2-}]_{cf}$ and $DIC_{cf}$ are approximately equal for all three formulations at the summertime maxima, but are lower during winter when using the $K_D$ formulations from Holcomb et al. (2016) or our Eq. (12), relative to McCulloch et al. (2017). Conversely, using the Allison (2017) $K_D$ formulation produces the opposite seasonal pattern with amplitude several times greater than the other $K_D$ formulations. This large discrepancy is not surprising given the behavior of the Allison (2017) $K_D$ formulation when retrospectively applied to the fluid composition of abiogenic aragonites (Figure 3).

## 6   A computer code for applying boron systematics to coral skeletons

We present here a user-friendly computer code for deriving $[CO_3^{2-}]_{cf}$ and $DIC_{cf}$ from boron systematics (supplemental files). The function is provided in both MATLAB and R formats, and it calculates $[CO_3^{2-}]_{cf}$ and $DIC_{cf}$ given inputs of $\delta^{11}B$, B/Ca, temperature, salinity, and water depth. It allows easy toggling between what we consider the three plausible $K_D$ formulations (Holcomb et al. (2016), McCulloch et al. (2017), and our new Eq. 12). Furthermore, the code permits a choice of $[B]_{sw}$
functions since Allison et al. (2014) and Allison (2017) used the relation between salinity and $[B]_{sw}$ from Uppstrom (1974), whereas D'Olivo and McCulloch (2017) and McCulloch et al. (2017) used that of Lee et al. (2010). The carbonate dissociation





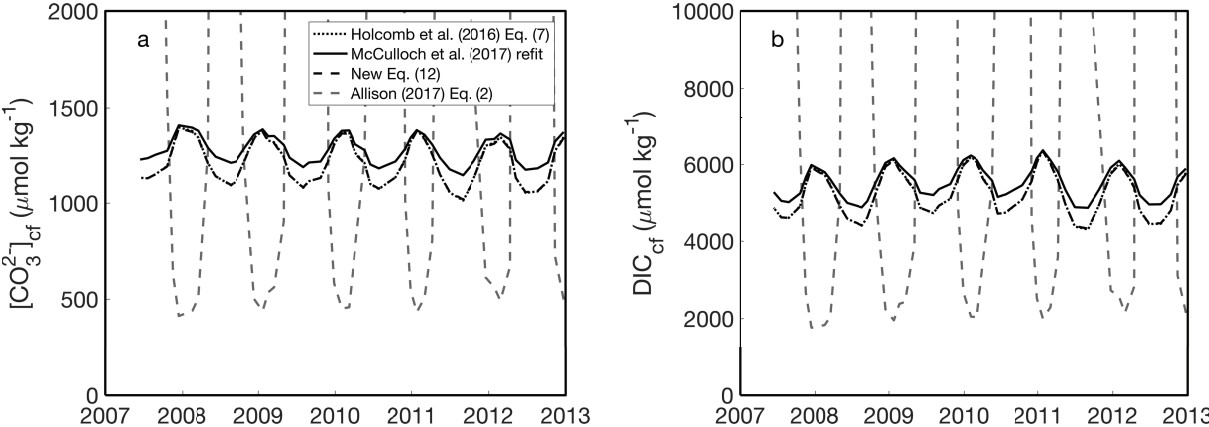

**Figure 6.** Application of the four $K_D$ formulations for the "Davies 2" *Porites* coral data from McCulloch et al. (2017). Derived $[CO_3^{2-}]_{cf}$ (a) and $DIC_{cf}$ (b) are plotted over multiple years using the $K_D$ formulations of Holcomb et al. (2016) (dotted black line), McCulloch et al. (2017) (solid black line), Eq. (12) (dashed black line), and Allison (2017) (dashed grey line).

constants can also be toggled between Dickson and Millero (1987) and Lueker et al. (2000). The code follows the calculations of CO2SYS (Lewis et al., 1998) for converting between pH scales and accounting for pressure effects on equilibrium constants, and uses the $\delta^{11}B_{sw}$ of Foster et al. (2010) and the $\alpha_{B3-B4}$ of Klochko et al. (2006).

Perhaps most importantly, the code propagates known uncertainties into the derivation of $[CO_3^{2-}]_{cf}$ and $DIC_{cf}$. These
uncertainties are estimated using a Monte Carlo scheme, in which random errors (assuming Gaussian distributions) are added to parameters while repeating the calculations many times. The non-systematic uncertainty of derived values depends on the measurement precisions of $\delta^{11}B$, B/Ca, temperature, and salinity. These will depend on the instruments and protocols used, and for $\delta^{11}B$ and B/Ca should be estimated by each laboratory, for example by repeated measurements of an external consistency standard. The systemic errors of derived values depend on the uncertainties of the various $K_D$ formulations, uncertainties
associated with $\delta^{11}B_{sw}$ (Foster et al., 2010), $[B]_{sw}$ (Lee et al., 2010), $\alpha_{B3-B4}$ (Klochko et al., 2006), and p$K_B$ (Dickson, 1990); and if known, any uncertainties in the accuracy of $\delta^{11}B$, B/Ca, temperature, and salinity measurements.

With our code, the parameter space of $[CO_3^{2-}]_{cf}$ derived from $\delta^{11}B$ and B/Ca, and the differences among $K_D$ formulations, can be readily visualized (Figure 7). This enables future applications of boron systematics to coral skeletons to consider how the choice of $K_D$ formulation affects the particular question being investigated. We also apply the code to calculate carbonate
system parameters using published $\delta^{11}B$ and B/Ca datasets (Figure 8). Interestingly, this analysis shows that coral calcifying fluid $[CO_3^{2-}]_{cf}$ and DIC are consistently positively correlated across studies (Figure 8b), whereas the sign of correlations between pH and both $[CO_3^{2-}]_{cf}$ and DIC varies (Figure 8c-d). Assuming $[CO_3^{2-}]_{cf}$ is the carbonate system parameter most important for aragonite precipitation, these patterns may suggest that elevating $DIC_{cf}$ is critical to the coral calcification process, although up-regulating pH is still important for shifting the carbonate system to favor $CO_3^{2-}$ over $HCO_3^-$.





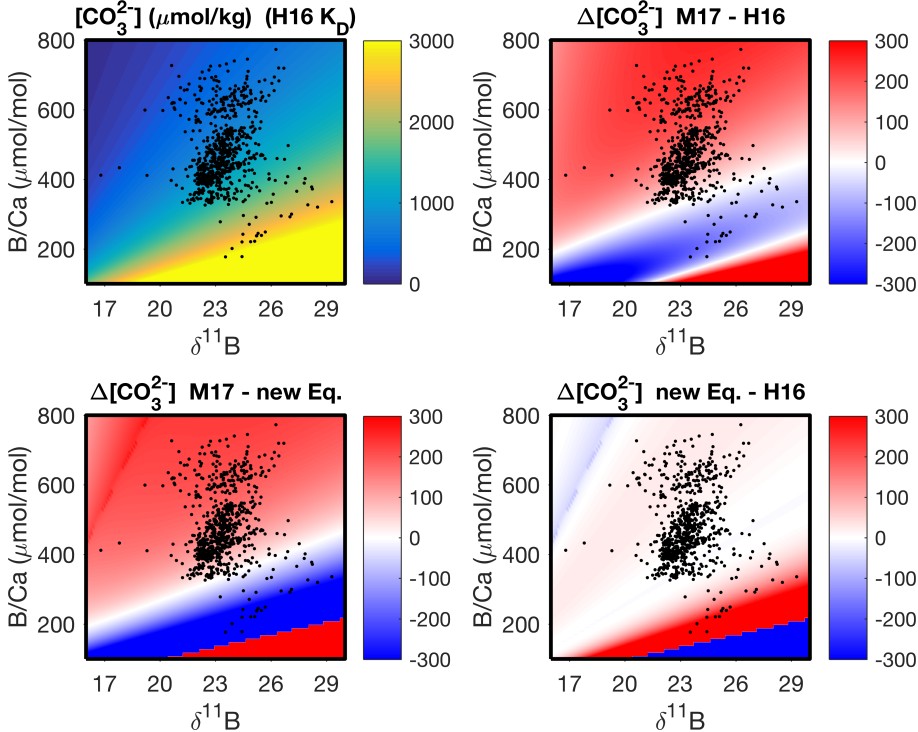

**Figure 7.** Application of our computer code to visualizing the parameter space of $[CO_3^{2-}]$ (in $\mu$mol kg$^{-1}$) derived from B/Ca and $\delta^{11}$B at 25 °C and salinity 35. The upper left panel shows absolute $[CO_3^{2-}]$ derived with the $K_D$ of Holcomb et al. (2016) ("H16"), whereas the other panels show the differences in $[CO_3^{2-}]$ between the $K_D$ formulations of H16, McCulloch et al. (2017) ("M17"), and our new Eq. (12). The black dots show coral data from the literature (see Figure 8 legend below). Note that the actual $[CO_3^{2-}]$ derived for the coral data will also depend on variations of the *in situ* temperature and salinity, which are not accounted for in the plots.

## 7 Which $K_D$ formulation to use?

Despite the availability of abiogenic B/Ca partitioning data from two experiments (Mavromatis et al., 2015; Holcomb et al., 2016), and several attempts to fit the data (Holcomb et al., 2016; Allison, 2017; McCulloch et al., 2017), it is important to recognize that uncertainties still remain, in particular an understanding of the controlling factors, and thus the appropriate
5   $K_D$ formulation to apply. From a mechanistic viewpoint, the key fundamental question that remains is whether the abiogenic data of Mavromatis et al. (2015) and Holcomb et al. (2016) are directly comparable and thus should be fit with a continuous function (*e.g.* Eq. 12), or if they are incomparable because Mavromatis et al. (2015) used NaCl solutions and Holcomb et al. (2016) used seawater. If they are comparable, then our new Eq. (12) or a similar fit to both datasets is the most appropriate $K_D$ formulation. Calcite precipitation studies provide some support for the hypothesis that crystal growth rate or $\Omega_{Ar}$ influences
10   B/Ca partitioning (Ruiz-Agudo et al., 2012; Uchikawa et al., 2015, 2017), but it is not yet known if these results can be extended to aragonite precipitation from seawater. Alternatively, if the solution chemistry makes the two experiments incomparable, the




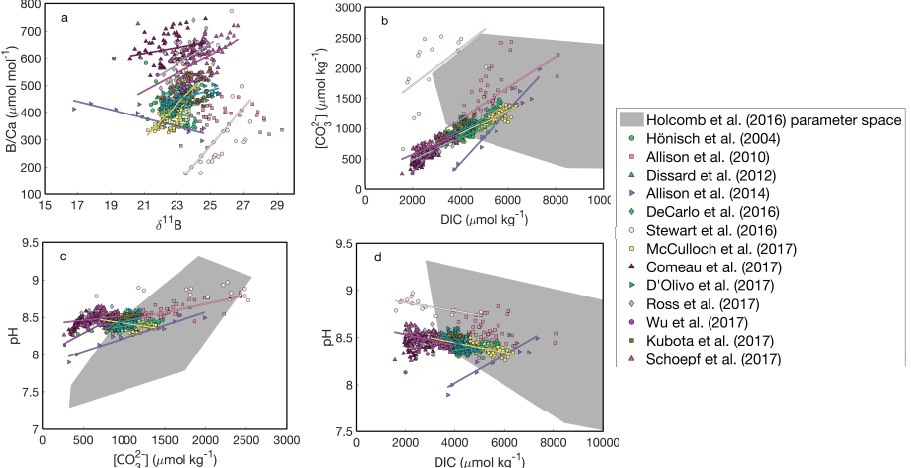

**Figure 8.** Correlations among coral calcifying fluid carbonate system parameters based on published boron systematics datasets: (a) B/Ca and $\delta^{11}$B, (b) [$CO_3^{2-}$] and DIC, (c) pH and [$CO_3^{2-}$], and (d) pH and DIC. Colors show different studies, and lines are plotted for significant (p < 0.05) correlations using all the data within each study. Grey area shows the convex hull of the parameter space covered in the abiogenic experiments of Holcomb et al. (2016). Calculations are performed using the Holcomb et al. (2016) $K_D$ formulation.

Holcomb et al. (2016) $K_D$ data are most likely the more suitable choice for corals because the experiments were conducted with seawater at comparable $\Omega_{Ar}$ to that of coral calcifying fluids (DeCarlo et al., 2017), and they can be fit as either a function of [$CO_3^{2-}$] or [$H^+$]. However, it is important to recognize that the parameter space of $CO_2$ system parameters covered in the Holcomb et al. (2016) experiments includes some, but not all, of the published coral data (Figure 8). Further, since we are
unable to conclusively distinguish whether the two abiogenic datasets are directly comparable, all three $K_D$ formulations may be considered equally valid until proven otherwise. Additional abiogenic experiments aimed at this question will clearly be useful in refining the boron systematics proxies.

From a practical standpoint, the $K_D$ formulations of Holcomb et al. (2016) and McCulloch et al. (2017) may be the most appropriate. Both produce unique solutions of [$CO_3^{2-}$] that increase with $\frac{[B(OH)_4^-]^{fluid}}{[B/Ca]^{aragonite}}$, and they effectively reconstruct fluid
[$CO_3^{2-}$] using the abiogenic aragonites precipitated from seawater. While our Eq. (12) produces [$CO_3^{2-}$]$_{cf}$ estimates that are nearly identical under most $\delta^{11}$B and B/Ca combinations to those derived using the Holcomb et al. (2016) $K_D$ formulation (Figure 7), Eq. (12) can have non-unique solutions, which could complicate interpretations of [$CO_3^{2-}$]$_{cf}$ in some cases.

A final consideration is that two of the $K_D$ formulations (Holcomb et al. (2016) and our new Eq. 12) are fit to [$CO_3^{2-}$]. Fitting Eq. (12) to a wider range of [$CO_3^{2-}$] helps to account for the different solution chemistries and associated growth rates
of the two abiogenic precipitation studies (Mavromatis et al., 2015; Holcomb et al., 2016), but $\Omega_{Ar}$ or crystal growth rate may be the true controlling factor (Watson, 2004; van der Weijden and van der Weijden, 2014). However, Holcomb et al. (2016) did not find a temperature dependence of B/Ca partitioning, as would be expected if precipitation rate influenced $K_D$. While growth rate is likely related to [$CO_3^{2-}$] (Burton and Walter, 1987), the two could decouple with changes in temperature or if coral calcifying fluid [$Ca^{2+}$]$_{cf}$ departs from seawater levels. Recent evidence combining Raman spectroscopy with boron



systematics suggests $[\text{Ca}^{2+}]_{cf}$ is within ~20% of seawater (DeCarlo et al., 2017), but this has yet to be tested on a range of coral species and locations. Thus, future abiogenic experiments designed to test under what conditions $[\text{CO}_3^{2-}]$ or crystal growth rates control B/Ca partitioning, as well as development of proxies for $[\text{Ca}^{2+}]_{cf}$, may improve the accuracy of deriving calcifying fluid carbonate chemistry from boron systematics.

**8    Conclusions**

Recent abiogenic aragonite precipitation experiments have made possible the application of boron systematics to quantifying the full carbonate system of coral calcifying fluid. However, a number of approaches to doing so have been utilized (Allison et al., 2014; Allison, 2017; D'Olivo and McCulloch, 2017; McCulloch et al., 2017), without a comprehensive analysis of which $K_D$ formulations are plausible (*i.e.* can reproduce the experimental fluid chemistry) or the implications for interpreting coral

skeletons. We evaluated four potential B/Ca $K_D$ formulations involving $\text{B(OH)}_4^-$ substituting for $\text{CO}_3^{2-}$ in the aragonite lattice. Our analysis suggests that there are at least three plausible $K_D$ formulations (Holcomb et al. (2016), McCulloch et al. (2017), and our new Eq. 12) that can be used to determine the $K_D$ and its dependence on fluid chemistry. Despite the differences in plausible approaches, we show that all three produce similar patterns in derived coral calcifying fluid carbonate chemistry. Nevertheless, subtle differences in derived carbonate chemistry remain among the approaches, and addressing these differences

should be the target of future abiogenic aragonite precipitation experiments. Finally, we present a code that computes coral calcifying fluid carbonate chemistry from boron systematics, and allows for comparison among different $K_D$ formulations.

*Code availability.*   Codes are available in the Supplement

**Appendix A**

In the main text, we used numerical solutions to predict $[\text{CO}_3^{2-}]$ based on Eq. (14). Here, we show an analytical solution to Eq.

(14) for the Allison (2017) $K_D$ formulation. Allison (2017) fit $K_D$ to $\Omega_{Ar}$ with a linear regression in the form:

$$K_D^{\text{B/Ca}} = a\Omega + b \tag{A1}$$

where

$$\Omega = \frac{[\text{CO}_3^{2-}][\text{Ca}^{2+}]}{K_{sp}} \tag{A2}$$

with concentrations in units of mol kg$^{-1}$ and where $K_{sp}$ is the solubility product. Inserting Eq. (A2) into Eq. (A1):

$$K_D^{\text{B/Ca}} = a\frac{[\text{CO}_3^{2-}][\text{Ca}^{2+}]}{K_{sp}} + b \tag{A3}$$

and then inserting Eq. (A3) into Eq. (14) of the main text:

$$[\text{CO}_3^{2-}] = (a\frac{[\text{CO}_3^{2-}][\text{Ca}^{2+}]}{K_{sp}} + b)\frac{[\text{B(OH)}_4^-]^{fluid}}{[\text{B/Ca}]^{aragonite}} \tag{A4}$$





where $[CO_3^{2-}]$ is in units of mol kg$^{-1}$. We must now solve Eq. (A4) for $[CO_3^{2-}]$. First, expand the right side of the equation:

$$[CO_3^{2-}] = a\frac{[CO_3^{2-}][Ca^{2+}]}{K_{sp}}\frac{[B(OH)_4^-]^{fluid}}{[B/Ca]^{aragonite}} + b\frac{[B(OH)_4^-]^{fluid}}{[B/Ca]^{aragonite}} \tag{A5}$$

Multiply both sides by $K_{sp}$:

$$[CO_3^{2-}]K_{sp} = a[CO_3^{2-}][Ca^{2+}]\frac{[B(OH)_4^-]^{fluid}}{[B/Ca]^{aragonite}} + b\frac{[B(OH)_4^-]^{fluid}}{[B/Ca]^{aragonite}}K_{sp} \tag{A6}$$

Collect all the $[CO_3^{2-}]$ terms on the left side of the equation:

$$[CO_3^{2-}]K_{sp} - a[CO_3^{2-}][Ca^{2+}]\frac{[B(OH)_4^-]^{fluid}}{[B/Ca]^{aragonite}} = b\frac{[B(OH)_4^-]^{fluid}}{[B/Ca]^{aragonite}}K_{sp} \tag{A7}$$

Factor out $[CO_3^{2-}]$:

$$[CO_3^{2-}]\left(K_{sp} - a[Ca^{2+}]\frac{[B(OH)_4^-]^{fluid}}{[B/Ca]^{aragonite}}\right) = b\frac{[B(OH)_4^-]^{fluid}}{[B/Ca]^{aragonite}}K_{sp} \tag{A8}$$

Solve for $[CO_3^{2-}]$:

$$[CO_3^{2-}] = \frac{b\frac{[B(OH)_4^-]^{fluid}}{[B/Ca]^{aragonite}}K_{sp}}{K_{sp} - a[Ca^{2+}]\frac{[B(OH)_4^-]^{fluid}}{[B/Ca]^{aragonite}}} \tag{A9}$$

In seawater at 25 °C and salinity 34, $[Ca^{2+}]$ is approximately 0.01 mol kg$^{-1}$ and $K_{sp}$ is $6.54x10^{-7}$ (Riley and Tongudai, 1967; Lewis et al., 1998). According to Allison (2017), $a$ is $1.48x10^{-4}$ and $b$ is $-1.30x10^{-4}$. Inserting these values in Eq. (A9):

$$[CO_3^{2-}] = \frac{(-1.30x10^{-4})\frac{[B(OH)_4^-]^{fluid}}{[B/Ca]^{aragonite}}(6.54x10^{-7})}{(6.54x10^{-7}) - (1.48x10^{-4})(0.01)\frac{[B(OH)_4^-]^{fluid}}{[B/Ca]^{aragonite}}} = \frac{-8.50x10^{-11}\frac{[B(OH)_4^-]^{fluid}}{[B/Ca]^{aragonite}}}{6.54x10^{-7} - 1.48x10^{-6}\frac{[B(OH)_4^-]^{fluid}}{[B/Ca]^{aragonite}}} \tag{A10}$$

The denominator equals zero (*i.e.* the solution is undefined) when $\frac{[B(OH)_4^-]^{fluid}}{[B/Ca]^{aragonite}} = \frac{6.54x10^{-7}}{1.48x10^{-6}} = 0.44$. If $\frac{[B(OH)_4^-]^{fluid}}{[B/Ca]^{aragonite}} < 0.44$,

then the denominator is positive, and since the numerator is always negative, this means that the predicted $[CO_3^{2-}]$ will be negative. Predicted $[CO_3^{2-}]$ will be highest when the denominator is a small negative number, which occurs when $\frac{[B(OH)_4^-]^{fluid}}{[B/Ca]^{aragonite}}$ is slightly greater than 0.44. As $\frac{[B(OH)_4^-]^{fluid}}{[B/Ca]^{aragonite}}$ increases » 0.44, the absolute value of the denominator increases more than that of the numerator because the coefficient attached to $\frac{[B(OH)_4^-]^{fluid}}{[B/Ca]^{aragonite}}$ is raised to the -6 power in the denominator and to the -11 power in the numerator. The implication is that predicted $[CO_3^{2-}]$ will decrease as $\frac{[B(OH)_4^-]^{fluid}}{[B/Ca]^{aragonite}}$ increases beyond 0.44. This is

the same conclusion reached in the main text, and is the opposite trend to that observed in the abiogenic aragonites (Figure 4).

*Competing interests.*  The authors declare no competing interests.

*Acknowledgements.*  The authors thank Glenn Gaetani for valuable comments. This study was funded by an ARC Laureate Fellowship (FL120100049) awarded to M.T.M., and the ARC Centre of Excellence for Coral Reef Studies (CE140100020).



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
