# Peer review of "Reviews and syntheses: Revisiting the boron systematics of aragonite and their application to coral calcification"

_Biogeosciences, 2018_

## Referee Comment (RC1) · Anonymous Referee #1 · 18 Feb 2018

I really enjoyed reading the manuscript. The authors summarized issues on the selection of Kd value (and its formula) and its potential influence on the calculation of full carbonate chemistry in the calcifying medium. The logic is concise, and I strongly recommend a publication of the manuscript.

The followings are my minor comments that may be helpful for the authors to improve the manuscript.

(pp. 2 Line 20–) I think almost nobody use stable carbon and oxygen isotopes as a proxy of carbonate chemistry, so you can delete the related sentences.

(pp. 7 Figure 2 and pp. 14 Figure 8) About pH and [H+]. I think [H+] presented in the

[Figure]

Figure 2 is that of solution used in the precipitation experiment. In Figure 8, on the other hand, they are calcifying fluid pH for coral data as well as solution pH for precipitation experiment. I would be better to clarify what each pH stand for in somewhere in the manuscript (in each figure caption?).

(pp. 10 Figure 4) Why do you use Kd value of 0.002 as an example of constant Kd?

(pp. 12 Figure 6) Is there any better way to plot these data? The difference between New Eq. (12) line and Allison (2017) line are very ambiguous.

(pp. 14 Line 17- pp. 15 Line 2) It is just a question. Is this the reason why you don't show a cross-plot of $\Omega$ar against the other parameters? (such as $\Omega$ar versus pH)

---

## Referee Comment (RC2) · Anonymous Referee #2 · 2 Mar 2018

DeCarlo et al. synthesize the (very recently developed) joint B/Ca-$\delta$11B system in aragonite corals as a proxy for coral calcifying fluid chemistry. Coral aragonite $\delta$11B has previously been applied as a calcifying fluid pH proxy, while recent studies of synthetic aragonite B/Ca suggest control by [CO32-]. If these results apply to corals, then coral aragonite B/Ca may reflect [CO32-] in the calcifying fluid. The ability to reconstruct [CO32-] (from B/Ca) and pH (from $\delta$11B) allows for solving the carbonate chemistry of coral calcifying fluid, which permits reconstructions of calcifying fluid DIC (among other parameters). This new approach hinges on the veracity of coral B/Ca to [CO32-

] reconstructions, which these authors test in detail. They also present a calcifying fluid calculation routine that propagates all uncertainties associated with the above calculations.

This is a nicely written and useful contribution, and I do support its publication, but I think it is missing one key component:

Primary concern/recommendation: Coral B/Ca as a [CO32-]cf proxy exploded in the last two years, in large part due to the works of these authors. While this contribution cites requisite previous reasoning (Holcomb et al. Chem Geol. 2016 for synthetic aragonite, and McCulloch et al. Nat. Comm. 2017), I do not find that the rationale for this approach has been sufficiently explored in previous publications. As the authors use this manuscript to comprehensively and quantitatively analyze KD formulations, I strongly encourage them to also take a step back and comprehensively evaluate the B/Ca-[CO32-] proxy system in corals and its inherent assumptions. Adding this to the quantitative treatment already provided would greatly enhance this contribution's readability and utility.

Guiding questions for this background: 1) What is known about patterns in coral B/Ca? How do features of these patterns (seasonal cycles, etc.) imply a relationship to [CO32-]cf and/or [DIC]cf? It seems previously published B/Ca data are already compiled in Figure 8, so this won't require much work.

2) What is known about coral [DIC]cf, both naturally and in controlled experiments (e.g., Cai et al., 2016; Comeau et al., 2017)? What are the limitations to direct measurements? (Schoepf et al. 2017 gave a nice overview of this, but I would appreciate seeing that reasoning here)

3) Two previous studies of paired foraminifera B/Ca and $\delta$11B concluded that joint reconstructions of [CO32-] and pH could not be used to reconstruct full ocean carbonate chemistry because the relative uncertainties in reconstructing Alk and DIC were larger than the entire range of these parameters in the modern ocean (Yu et al., EPSL 2010;

Rae et al., EPSL 2011). What is different in corals that make this application feasible? I think it probably relates to the much bigger ranges of [CO32-] and/or [DIC] in coral calcifying fluids vs. seawater, but I'd like to hear that from the authors. In general, the coral joint B/Ca and $\delta$11B approach needs to be presented within the context of previous (unsuccessful) open ocean efforts.

Specific comments: Page 3, L1 (relevant for Section 2): For most boron proxy applications, inorganic carbonate precipitation experiments do not reflect biogenic carbonates as well as our community would like (see, e.g., Allen and Hönisch, 2011; 2012; Uchikawa et al. 2015, 2017, review in Rae and Foster, 2016; Rae 2018 book chapters). Please defend why applying a KD derived from synthetic aragonite B/Ca is appropriate for coral aragonite in light of issues observed in other boron applications. This discussion could fit well in Section 7 (p. 13).

Page 5, L31: What might compositional effects on B/Ca partitioning look like? This is a critical point for two reasons: 1) If compositional effects do exist, then B/Ca partitioning is not effectively described by KD, and instead requires additional parameters related to varying solution chemistry than only [CO32-] and [Ca2+]. 2) If compositional effects do exist, then application of B/Ca-[CO32-] approach to coral calcifying fluid would carry additional uncertainty because the calcifying fluid composition is not unaltered seawater (because of ion pumps such as Ca-ATPase) Note: I feel that the authors nicely dealt with comparing the B/Ca data from Mavromatis and Holcomb nicely throughout the manuscript, and their approach of using both datasets to define KD in terms of CO32- (Equation 12) implies that compositional effects do not matter. But I think it is important for them to note that compositional effects could undermine the application of the B/Ca-[CO32-] approach to non-seawater media (which includes the calcifying fluid).

Page 12, L14-19 and Figure 8: Suggest you change the order of figures, starting from the measured parameters ($\delta$11B and B/Ca, in a), then each converted to their independent parameters (pH and CO32-), and finally plots vs. DIC, which requires both

parameters. It is tough to say whether the correlation between DIC and CO32- is "interesting" or even surprising, because the calculation of DIC depends on pH and CO32-. Because pH and DIC do not correlate well, changes in DIC are probably principally driven by changes in [CO32-] (and hence coral B/Ca). This could be worth exploring with a sensitivity test.

Page 14, L16-17: In section 2, the authors state that Holcomb et al. (2016) only performed two experiments at each offset temperature, and that this was insufficient to quantify temperature effects on precipitation rate. Are the data also too limited to find a temperature dependence on B/Ca partitioning?

Figure comments. Please label panels a) through d) (or however many panels) in each figure (some are missing). I would recommend increasing the font size of these labels; they are difficult to see.

Figure 6. I only see three line types on here (solid-McCulloch, gray dash-Allison, and then a dot dash that may be both the Holcomb and Equation 12 lines?) If the Holcomb and Equation 12 lines fall on top of each other, please say so in the text and figure caption. Additionally, while the authors MATLAB routine calculates a propagated uncertainty on derived [CO32-]cf and [DIC]cf, no uncertainities are plotted. Please illustrate this uncertainty on Figure 6. How does the propagated uncertainty affect the conclusion about applicability of McCulloch, Holcomb, and Equation 12 lines? Are they truly any different from each other (tested statistically)?

Figure 7. Panel labeling. Also, do not use $\Delta$[CO32-] in titles, as this is a well-used carbonate chemistry term. Suggest changing titles to "[CO32-]cf difference" or "[CO32-]cf M17 – [CO32-]cf H16". Please specify that [CO32-] is [CO32-]cf on figures and in caption. Finally, the color schemes are a bit tough to follow. In b) through d), white is good, right?

Figure 8. Tough figure to read—recommend brighter symbol colors and making the gray shading for the Holcomb et al. data lighter.

---

## Author Comment (AC1) · 6 Apr 2018

I really enjoyed reading the manuscript. The authors summarized issues on the selection of Kd value (and its formula) and its potential influence on the calculation of full carbonate chemistry in the calcifying medium. The logic is concise, and I strongly recommend a publication of the manuscript.

The followings are my minor comments that may be helpful for the authors to improve the manuscript.

(pp. 2 Line 20–) I think almost nobody use stable carbon and oxygen isotopes as a

proxy of carbonate chemistry, so you can delete the related sentences.

**We agree with the reviewer that carbon and oxygen isotope ratios are not commonly applied as carbonate system proxies in corals. This phrasing will be revised to indicate that they are not typically applied in this way, but they are theoretically sensitive to carbonate chemistry. We prefer to still mention carbon and oxygen isotopes because they are examples of geochemical proxies that are sensitive to the carbonate system, yet are not very useful proxies in corals due to a variety of vital effects.**

(pp. 7 Figure 2 and pp. 14 Figure 8) About pH and [H+]. I think [H+] presented in the Figure 2 is that of solution used in the precipitation experiment. In Figure 8, on the other hand, they are calcifying fluid pH for coral data as well as solution pH for precipitation experiment. I would be better to clarify what each pH stand for in somewhere in the manuscript (in each figure caption?).

**We will make clear the distinction between coral calcifying fluid pH (or H+) and the abiogenic experimental fluid pH, both in the captions and axis labels.**

(pp. 10 Figure 4) Why do you use Kd value of 0.002 as an example of constant Kd?

**The value of 0.002 was selected simply as an example that intersects the abiogenic data near the range of [$CO_3^{2-}$] found in corals. We could choose any other value, which would be a similar line but further from the abiogenic dataset. The main message is that the constant Kd underestimates the sensitivity of [$CO_3^{2-}$] to borate/(B/Ca), which is made clear by the best-case example with Kd of 0.002.**

(pp. 12 Figure 6) Is there any better way to plot these data? The difference between New Eq. (12) line and Allison (2017) line are very ambiguous.

**We agree that the lines are very close together. We will revise this figure to show a narrower y-axis that will enable better visualization of the different lines.**

(pp. 14 Line 17- pp. 15 Line 2) It is just a question. Is this the reason why you don't

show a cross-plot of $\delta^{11}$ar against the other parameters? (such as $\delta^{11}$ar versus pH)

**Yes, we prefer to plot only boron-derived [CO$_3^{2-}$], rather than saturation state, because boron systematics really only provide information regarding pH and [CO$_3^{2-}$], not [Ca$^{2+}$].**

---

## Author Comment (AC2) · 6 Apr 2018

DeCarlo et al. synthesize the (very recently developed) joint B/Ca-$\delta^{11}$B system in aragonite corals as a proxy for coral calcifying fluid chemistry. Coral aragonite $\delta^{11}$B has previously been applied as a calcifying fluid pH proxy, while recent studies of synthetic aragonite B/Ca suggest control by [$CO_3^{2-}$]. If these results apply to corals, then coral aragonite B/Ca may reflect [CO32-] in the calcifying fluid. The ability to reconstruct [$CO_3^{2-}$] (from B/Ca) and pH (from $\delta^{11}$B) allows for solving the carbonate chemistry of coral calcifying fluid, which permits reconstructions of calcifying fluid DIC (among other parameters). This new approach hinges on the veracity of coral B/Ca to [$CO_3^{2-}$] reconstructions, which these authors test in detail. They also present a calcifying fluid calculation routine that propagates all uncertainties associated with the above calculations.

This is a nicely written and useful contribution, and I do support its publication, but I think it is missing one key component:

Primary concern/recommendation: Coral B/Ca as a $[CO_3^{2-}]_{cf}$ proxy exploded in the last two years, in large part due to the works of these authors. While this contribution cites requisite previous reasoning (Holcomb et al. Chem Geol. 2016 for synthetic aragonite, and McCulloch et al. Nat. Comm. 2017), I do not find that the rationale for this approach has been sufficiently explored in previous publications. As the authors use this manuscript to comprehensively and quantitatively analyze KD formulations, I strongly encourage them to also take a step back and comprehensively evaluate the B/Ca-$[CO_3^{2-}]$ proxy system in corals and its inherent assumptions. Adding this to the quantitative treatment already provided would greatly enhance this contribution's readability and utility.

Guiding questions for this background: 1) What is known about patterns in coral B/Ca? How do features of these patterns (seasonal cycles, etc.) imply a relationship to $[CO_3^{2-}]_{cf}$ and/or $[DIC]_{cf}$? It seems previously published B/Ca data are already compiled in Figure 8, so this won't require much work.

**We will add some discussion on patterns of B/Ca in coral skeletons. However, it is difficult to interpret B/Ca alone because it is not directly related to $[CO_3^{2-}]$, but rather depends also on borate concentration (i.e. B/Ca depends on both pH and $[CO_3^{2-}]$). Nevertheless, we will add more acknowledgement of previous studies of coral B/Ca ratios.**

2) What is known about coral $[DIC]_{cf}$, both naturally and in controlled experiments (e.g., Cai et al., 2016; Comeau et al., 2017)? What are the limitations to direct measurements? (Schoepf et al. 2017 gave a nice overview of this, but I would appreciate

seeing that reasoning here)

**We will add a discussion on calcifying fluid DIC. There is a substantial, and currently unresolved, difference between DIC derived from boron systematics ($DIC_{cf}$ > seawater) and from microsensors ($DIC_{cf}$ < seawater). We will discuss potential reasons for this difference, and the implications for understanding coral calcification.**

3) Two previous studies of paired foraminifera B/Ca and $\delta^{11}B$ concluded that joint reconstructions of $[CO_3^{2-}]$ and pH could not be used to reconstruct full ocean carbonate chemistry because the relative uncertainties in reconstructing Alk and DIC were larger than the entire range of these parameters in the modern ocean (Yu et al., EPSL 2010; Rae et al., EPSL 2011). What is different in corals that make this application feasible? I think it probably relates to the much bigger ranges of $[CO_3^{2-}]$ and/or [DIC] in coral calcifying fluids vs. seawater, but I'd like to hear that from the authors. In general, the coral joint B/Ca and $\delta^{11}B$ approach needs to be presented within the context of previous (unsuccessful) open ocean efforts.

**We will add a discussion of applying boron systematics to reconstruct seawater chemistry. Like the foraminifera studies mentioned, efforts to reconstruct ocean carbonate chemistry with corals are not very successful because the changes within the calcifying fluid often far exceed natural variability of seawater. Thus, while boron systematics is a useful tool for understanding coral calcification and its sensitivity to changes in reef environments, it may not be generally applicable for deriving ocean chemistry. We will make this point clear in the revised manuscript.**

Specific comments:

Page 3, L1 (relevant for Section 2): For most boron proxy applications, inorganic carbonate precipitation experiments do not reflect biogenic carbonates as well as our community would like (see, e.g., Allen and Hönisch, 2011; 2012; Uchikawa et al. 2015,

2017, review in Rae and Foster, 2016; Rae 2018 book chapters). Please defend why applying a KD derived from synthetic aragonite B/Ca is appropriate for coral aragonite in light of issues observed in other boron applications. This discussion could fit well in Section 7 (p. 13).

**We will add discussion of this topic. In general, it is difficult to validate the application of Kd derived in abiogenic experiments to coral skeletons because independent data of coral calcifying fluid chemistry are scarce. Microsensor and fluorescent dye measurements of calcifying fluid pH are broadly similar to boron isotope-derived pH, but the one study of calcifying fluid DIC derived from microsensors differs from boron systematics results. However, boron systematics are broadly similar with constraints from U/Ca and Raman spectroscopy, which we will add to the revised manuscript.**

Page 5, L31: What might compositional effects on B/Ca partitioning look like? This is a critical point for two reasons: 1) If compositional effects do exist, then B/Ca partitioning is not effectively described by KD, and instead requires additional parameters related to varying solution chemistry than only $[CO_3^{2-}]$ and $[Ca^{2+}]$. 2) If compositional effects do exist, then application of B/Ca-$[CO_3^{2-}]$ approach to coral calcifying fluid would carry additional uncertainty because the calcifying fluid composition is not unaltered seawater (because of ion pumps such as Ca-ATPase) Note: I feel that the authors nicely dealt with comparing the B/Ca data from Mavromatis and Holcomb nicely throughout the manuscript, and their approach of using both datasets to define KD in terms of CO32- (Equation 12) implies that compositional effects do not matter. But I think it is important for them to note that compositional effects could undermine the application of the B/Ca-$[CO_3^{2-}]$ approach to non-seawater media (which includes the calcifying fluid).

**The reviewer makes a good point here. It is important to note that the Holcomb et al. (2016) experiments include a range of seawater chemical manipulations, including [Mg], [Ca], and [Sr] exceeding changes typically thought to occur within the calcifying fluid, without clear effects on Kd. Thus, we do not think there are**

**strong sensitivities of Kd to trace element variations. Yet it is possible that there are subtle effects, which are not apparent in Holcomb et al. (2016) because the fluids are all broadly similar to seawater, but do become apparent in Mavromatis et al. (2015) since the fluid chemistry departs substantially from seawater for many elements. We will discuss this issue further in the revised manuscript.**

Page 12, L14-19 and Figure 8: Suggest you change the order of figures, starting from the measured parameters ($\delta^{11}$B and B/Ca, in a), then each converted to their independent parameters (pH and $CO_3^{2-}$), and finally plots vs. DIC, which requires both parameters. It is tough to say whether the correlation between DIC and $CO_3^{2-}$ is "interesting" or even surprising, because the calculation of DIC depends on pH and $CO_3^{2-}$. Because pH and DIC do not correlate well, changes in DIC are probably principally driven by changes in [$CO_3^{2-}$] (and hence coral B/Ca). This could be worth exploring with a sensitivity test.

**We agree with this suggestion, and we will revise the order of panels in Figure 8. In terms of deriving DIC, yes it appears to depend most strongly on [$CO_3^{2-}$]. However, in terms of modification within the calcifying fluid, it may be that $CO_2$ diffusion drives DIC changes, which in turn affect [$CO_3^{2-}$].**

Page 14, L16-17: In section 2, the authors state that Holcomb et al. (2016) only performed two experiments at each offset temperature, and that this was insufficient to quantify temperature effects on precipitation rate. Are the data also too limited to find a temperature dependence on B/Ca partitioning?

**We will revise the statement regarding the quantification of temperature effects on precipitation rate. The Holcomb et al. (2016) data are generally consistent with Burton and Walter (1987) in that precipitation rate increases with temperature, and the data are sufficient to demonstrate this. However, Burton and Walter (1987) show that the order of the reaction changes with temperature, which requires a full calibration dataset (i.e. more than 2 experiments) for each tem-**

**perature. Thus, in the revised manuscript we will describe that we do see a temperature dependence of reaction rate, but that we cannot go as far as Burton and Walter (1987) in quantifying changes in the reaction order. Holcomb et al. (2016) already reported that there was no apparent temperature effect on B/Ca partitioning between 20 and 40 °C.**

Figure comments.

Please label panels a) through d) (or however many panels) in each figure (some are missing). I would recommend increasing the font size of these labels; they are difficult to see.

**We will add panel labels to all figures.**

Figure 6. I only see three line types on here (solid-McCulloch, gray dash-Allison, and then a dot dash that may be both the Holcomb and Equation 12 lines?) If the Holcomb and Equation 12 lines fall on top of each other, please say so in the text and figure caption. Additionally, while the authors MATLAB routine calculates a propagated uncertainty on derived $[CO_3^{2-}]_{cf}$ and $[DIC]_{cf}$, no uncertainities are plotted. Please illustrate this uncertainty on Figure 6. How does the propagated uncertainty affect the conclusion about applicability of McCulloch, Holcomb, and Equation 12 lines? Are they truly any different from each other (tested statistically)?

**We will revise Figure 6 to more clearly show the separate lines, and we will include error bars.**

Figure 7. Panel labeling. Also, do not use $\Delta[CO_3^{2-}]_{cf}$ in titles, as this is a well-used carbonate chemistry term. Suggest changing titles to "$[CO_3^{2-}]_{cf}$ difference" or "$[CO_3^{2-}]_{cf}$ M17 – $[CO_3^{2-}]_{cf}$ H16". Please specify that $[CO_3^{2-}]$ is $[CO_3^{2-}]_{cf}$ on figures and in caption. Finally, the color schemes are a bit tough to follow. In b) through d), white is good, right?

**We will change the panel titles as suggested. We prefer to keep the color scheme**

**as it is a common (and we believe effective) way to visualize anomalies because it is easy to see where the two formulas are consistent (white) or one higher than the other (red or blue).**

Figure 8. Tough figure to read, recommend brighter symbol colors and making the ËĞ gray shading for the Holcomb et al. data lighter.

**We will make the symbol colors clearer and the gray shading lighter.**

---

## Author Response (AR1)

Dear Editor,

Thank you for handling our manuscript, "Revisiting the boron systematics of aragonite and their application to coral calcification" (bg-2018-77). Two reviewers provided constructive comments that highlighted several areas requiring clarification or additional discussion. We have revised the manuscript following these helpful suggestions, which have improved the quality of our manuscript. Below, we respond (**in bold**) to each reviewer comment (plain text), with the modified text highlighted in yellow throughout the manuscript.

We greatly appreciate the time you have devoted to our manuscript as editor, and we look forward to publishing in *Biogeosciences*.

Sincerely,
Thomas DeCarlo, Michael Holcomb, and Malcolm McCulloch

**Responses to reviewer comments:**

Reviewer 1:

I really enjoyed reading the manuscript. The authors summarized issues on the selection of Kd value (and its formula) and its potential influence on the calculation of full carbonate chemistry in the calcifying medium. The logic is concise, and I strongly recommend a publication of the manuscript.

The followings are my minor comments that may be helpful for the authors to improve the manuscript.

(pp. 2 Line 20–) I think almost nobody use stable carbon and oxygen isotopes as a proxy of carbonate chemistry, so you can delete the related sentences.

**We agree with the reviewer that carbon and oxygen isotope ratios are not commonly applied as carbonate system proxies in corals. This phrasing has been revised to indicate that they are not typically applied in this way (page 2, lines 21-24), but they are theoretically sensitive to carbonate chemistry. We prefer to still mention carbon and oxygen isotopes because they are examples of geochemical proxies that are sensitive to the carbonate system, yet are not very useful proxies in corals due to a variety of vital effects.**

(pp. 7 Figure 2 and pp. 14 Figure 8) About pH and [H+]. I think [H+] presented in the Figure 2 is that of solution used in the precipitation experiment. In Figure 8, on the other hand, they are calcifying fluid pH for coral data as well as solution pH for precipitation experiment. I would be better to clarify what each pH stand for in somewhere in the manuscript (in each figure caption?).

**We have made clear the distinction between coral calcifying fluid pH (or $H^+$) and the abiogenic experimental fluid pH, both in the captions and axis labels (changes to axes made in Figure 2 and Figure 8).**

(pp. 10 Figure 4) Why do you use Kd value of 0.002 as an example of constant Kd?

**The value of 0.002 was selected simply as an example that intersects the abiogenic data near the range of $[CO_3^{2-}]$ found in corals (added to page 10, lines 15-16). We could choose any other value, which would be a similar line but further from the abiogenic dataset. The main message is that the constant Kd underestimates the sensitivity of $[CO_3^{2-}]$ to borate/(B/Ca), which is made clear by the best-case example with Kd of 0.002.**

(pp. 12 Figure 6) Is there any better way to plot these data? The difference between New Eq. (12) line and Allison (2017) line are very ambiguous.

**We have revised Figure 6 in several ways: Firstly, we now plot only $[CO_3^{2-}]$ (and not DIC) because the purpose of the plot is to demonstrate differences in derived $[CO_3^{2-}]$ among Kd formulations, and the derived DIC follows the same pattern. This enables the $[CO_3^{2-}]$ plot to be larger and thus more clearly visualized. Secondly, we made two separate $[CO_3^{2-}]$ axes, which allows us to focus more closely on the small differences between the Holcomb et al. (2016), McCulloch et al. (2017), and Eq. (12) formulations. Finally, we added error bars to the lines.**

(pp. 14 Line 17- pp. 15 Line 2) It is just a question. Is this the reason why you don't show a cross-plot of Ωar against the other parameters? (such as Ωar versus pH)

**Yes, we prefer to plot only boron-derived $[CO_3^{2-}]$, rather than saturation state, because boron systematics really only provide information regarding pH and $[CO_3^{2-}]$, not $[Ca^{2+}]$.**

Reviewer 2:

DeCarlo et al. synthesize the (very recently developed) joint B/Ca-δ11B system in aragonite corals as a proxy for coral calcifying fluid chemistry. Coral aragonite δ11B has previously been applied as a calcifying fluid pH proxy, while recent studies of synthetic aragonite B/Ca suggest control by [CO32-]. If these results apply to corals, then coral aragonite B/Ca may reflect [CO32-] in the calcifying fluid. The ability to reconstruct [CO32-] (from B/Ca) and pH (from δ11B) allows for solving the carbonate chemistry of coral calcifying fluid, which permits reconstructions of calcifying fluid DIC (among other parameters). This new approach hinges on the veracity of coral B/Ca to [CO32-] reconstructions, which these authors test in detail. They also present a calcifying fluid calculation routine that propagates all uncertainties associated with the above calculations.

This is a nicely written and useful contribution, and I do support its publication, but I think it is missing one key component:

Primary concern/recommendation: Coral B/Ca as a [CO32-]cf proxy exploded in the last two years, in large part due to the works of these authors. While this contribution cites requisite previous reasoning (Holcomb et al. Chem Geol. 2016 for synthetic aragonite, and McCulloch et al. Nat. Comm. 2017), I do not find that the rationale for this approach has been sufficiently explored in previous publications. As the authors use this manuscript to comprehensively and quantitatively analyze KD formulations, I strongly encourage them to

also take a step back and comprehensively evaluate the B/Ca-[CO32-] proxy system in corals and its inherent assumptions. Adding this to the quantitative treatment already provided would greatly enhance this contribution's readability and utility.

Guiding questions for this background:
1) What is known about patterns in coral B/Ca? How do features of these patterns (seasonal cycles, etc.) imply a relationship to [CO32- ]cf and/or [DIC]cf? It seems previously published B/Ca data are already compiled in Figure 8, so this won't require much work.

**We added some discussion on patterns of B/Ca in coral skeletons (e.g. acknowledging previous reports of seasonality on page 11, lines 12-14). However, it is difficult to interpret B/Ca alone because it is not directly related to $[CO_3^{2-}]$, but rather depends also on borate concentration (*i.e.* B/Ca depends on both pH and $[CO_3^{2-}]$), which we now highlight on page 13, lines 5-8. Additionally, we state directly that our focus for this study is on the combined application of B/Ca and $\delta^{11}B$ (page 3, lines 16-17).**

2) What is known about coral [DIC]cf, both naturally and in controlled experiments (e.g., Cai et al., 2016; Comeau et al., 2017)? What are the limitations to direct measurements? (Schoepf et al. 2017 gave a nice overview of this, but I would appreciate seeing that reasoning here)

**We added an extended discussion of calcifying fluid DIC (page 13, lines 18-30). Here, we discuss that there is a substantial, and currently unresolved, difference between DIC derived from boron systematics ($DIC_{cf}$ > seawater) and from microsensors ($DIC_{cf}$ < seawater). Additionally, we describe the implications for understanding coral calcification, and acknowledge that there is some independent supporting evidence for the high $DIC_{cf}$ scenario (page 13, lines 24-28).**

3) Two previous studies of paired foraminifera B/Ca and δ11B concluded that joint reconstructions of [CO32-] and pH could not be used to reconstruct full ocean carbonate chemistry because the relative uncertainties in reconstructing Alk and DIC were larger than the entire range of these parameters in the modern ocean (Yu et al., EPSL 2010; Rae et al., EPSL 2011). What is different in corals that make this application feasible? I think it probably relates to the much bigger ranges of [CO32-] and/or [DIC] in coral calcifying fluids vs. seawater, but I'd like to hear that from the authors. In general, the coral joint B/Ca and δ11B approach needs to be presented within the context of previous (unsuccessful) open ocean efforts.

**We added a discussion of the difficulty in applying boron systematics to reconstruct seawater chemistry (page 13, lines 13-17). Like the foraminifera studies mentioned by the reviewer, efforts to reconstruct ocean carbonate chemistry with corals are not very successful because the changes within the calcifying fluid often far exceed natural variability of seawater. Thus, while boron systematics is a useful tool for understanding coral calcification and its sensitivity to changes in reef environments, it may not be generally applicable for deriving ocean chemistry.**

Specific comments:

Page 3, L1 (relevant for Section 2): For most boron proxy applications, inorganic carbonate precipitation experiments do not reflect biogenic carbonates as well as our community would

like (see, e.g., Allen and Hönisch, 2011; 2012; Uchikawa et al. 2015, 2017, review in Rae and Foster, 2016; Rae 2018 book chapters). Please defend why applying a KD derived from synthetic aragonite B/Ca is appropriate for coral aragonite in light of issues observed in other boron applications. This discussion could fit well in Section 7 (p. 13).

**In general, it is difficult to validate the application of Kd derived in abiogenic experiments to coral skeletons because independent data of coral calcifying fluid chemistry are scarce. Microsensor and fluorescent dye measurements of calcifying fluid pH are broadly similar to boron isotope-derived pH, but the one study of calcifying fluid DIC derived from microsensors differs from boron systematics results. However, boron systematics are broadly similar with constraints from U/Ca and Raman spectroscropy, which we have added to the manuscript (page 13, lines 24-28).**

Page 5, L31: What might compositional effects on B/Ca partitioning look like? This is a critical point for two reasons: 1) If compositional effects do exist, then B/Ca partitioning is not effectively described by KD, and instead requires additional parameters related to varying solution chemistry than only [CO32-] and [Ca2+]. 2) If compositional effects do exist, then application of B/Ca-[CO32-] approach to coral calcifying fluid would carry additional uncertainty because the calcifying fluid composition is not unaltered seawater (because of ion pumps such as Ca-ATPase) Note: I feel that the authors nicely dealt with comparing the B/Ca data from Mavromatis and Holcomb nicely throughout the manuscript, and their approach of using both datasets to define KD in terms of CO32- (Equation 12) implies that compositional effects do not matter. But I think it is important for them to note that compositional effects could undermine the application of the B/Ca-[CO32-] approach to non-seawater media (which includes the calcifying fluid).

**The reviewer makes a good point here. It is important to note that the Holcomb et al. (2016) experiments include a range of seawater chemical manipulations, including [Mg], [Ca], and [Sr] exceeding changes typically thought to occur within the calcifying fluid, without clear effects on Kd. Thus, we do not think there are strong sensitivities of Kd to trace element variations. Yet it is possible that there are subtle effects, which are not apparent in Holcomb et al. (2016) because the fluids are all broadly similar to seawater, but do become apparent in Mavromatis et al. (2015) since the fluid chemistry departs substantially from seawater for many elements. We have added this discussion to page 6, lines 1-11.**

Page 12, L14-19 and Figure 8: Suggest you change the order of figures, starting from the measured parameters (δ11B and B/Ca, in a), then each converted to their independent parameters (pH and CO32-), and finally plots vs. DIC, which requires both parameters. It is tough to say whether the correlation between DIC and CO32- is "interesting" or even surprising, because the calculation of DIC depends on pH and CO32-. Because pH and DIC do not correlate well, changes in DIC are probably principally driven by changes in [CO32-] (and hence coral B/Ca). This could be worth exploring with a sensitivity test.

**We agree with this suggestion, and we have revised the order of panels in Figure 8 as suggested. In terms of deriving DIC, yes it appears to depend most strongly on $[CO_3^{2-}]$. However, in terms of modification within the calcifying fluid, it may be that $CO_2$ diffusion drives DIC changes, which in turn affect $[CO_3^{2-}]$.**

Page 14, L16-17: In section 2, the authors state that Holcomb et al. (2016) only performed two experiments at each offset temperature, and that this was insufficient to quantify temperature effects on precipitation rate. Are the data also too limited to find a temperature dependence on B/Ca partitioning?

**We revised the statement regarding the quantification of temperature effects on precipitation rate (page 5, lines 12-15). The Holcomb et al. (2016) data are generally consistent with Burton and Walter (1987) in that precipitation rate increases with temperature, and the data are sufficient to demonstrate this. However, Burton and Walter (1987) show that the order of the reaction changes with temperature, which requires a full calibration dataset (i.e. more than 2 experiments) for each temperature. Thus, we now describe that we do see a temperature dependence of reaction rate, but that we cannot go as far as Burton and Walter (1987) in quantifying changes in the reaction order. In addition, Holcomb et al. (2016) already reported that there was no apparent temperature effect on B/Ca partitioning between 20 and 40 °C (page 16, lines 6-7).**

Figure comments.

Please label panels a) through d) (or however many panels) in each figure (some are missing). I would recommend increasing the font size of these labels; they are difficult to see.

**We added panel labels to all figures (Figure 7 has been revised).**

Figure 6. I only see three line types on here (solid-McCulloch, gray dash-Allison, and then a dot dash that may be both the Holcomb and Equation 12 lines?) If the Holcomb and Equation 12 lines fall on top of each other, please say so in the text and figure caption. Additionally, while the authors MATLAB routine calculates a propagated uncertainty on derived [CO32-]cf and [DIC]cf, no uncertainities are plotted. Please illustrate this uncertainty on Figure 6. How does the propagated uncertainty affect the conclusion about applicability of McCulloch, Holcomb, and Equation 12 lines? Are they truly any different from each other (tested statistically)?

**As described above in response to comments from Reviewer 1, we revised Figure 6 to more clearly show the separate lines and we included error bars.**

Figure 7. Panel labeling. Also, do not use Δ[CO32-] in titles, as this is a well-used carbonate chemistry term. Suggest changing titles to "[CO32-]cf difference" or "[CO32- ]cf M17 – [CO32-]cf H16". Please specify that [CO32-] is [CO32-]cf on figures and in caption. Finally, the color schemes are a bit tough to follow. In b) through d), white is good, right?

**We changed the panel titles as suggested (Figure 7). We prefer to keep the color scheme as it is a common (and we believe effective) way to visualize anomalies because it is easy to see where the two formulas are consistent (white) or one higher than the other (red or blue).**

Figure 8. Tough figure to read, recommend brighter symbol colors and making the ˘ gray shading for the Holcomb et al. data lighter.

**We made the symbol colors clearer and the gray shading lighter in Figure 8.**

[revised manuscript text omitted]